# Xanthine Oxidoreductase-Mediated Superoxide Production Is Not Involved in the Age-Related Pathologies in *Sod1*-Deficient Mice

**DOI:** 10.3390/ijms22073542

**Published:** 2021-03-29

**Authors:** Shuichi Shibuya, Kenji Watanabe, Yusuke Ozawa, Takahiko Shimizu

**Affiliations:** 1Aging Stress Response Research Project Team, National Center for Geriatrics and Gerontology, Obu, Aichi 474-8511, Japan; s-shibuya@ncgg.go.jp (S.S.); kng-wtnb@ncgg.go.jp (K.W.); 2Department of Endocrinology, Hematology, and Geriatrics, Chiba University Graduate School of Medicine, Chiba, Chiba 260-8670, Japan; ozawayusuke3@gmail.com

**Keywords:** superoxide, SOD1, xanthine oxidoreductase, aging, oxidative stress

## Abstract

Reactive oxygen species (ROS) metabolism is regulated by the oxygen-mediated enzyme reaction and antioxidant mechanism within cells under physiological conditions. Xanthine oxidoreductase (XOR) exhibits two inter-convertible forms (xanthine oxidase (XO) and xanthine dehydrogenase (XDH)), depending on the substrates. XO uses oxygen as a substrate and generates superoxide (O_2_^•−^) in the catalytic pathway of hypoxanthine. We previously showed that superoxide dismutase 1 (SOD1) loss induced various aging-like pathologies via oxidative damage due to the accumulation of O_2_^•−^ in mice. However, the pathological contribution of XO-derived O_2_^•−^ production to aging-like tissue damage induced by SOD1 loss remains unclear. To investigate the pathological significance of O_2_^•−^ derived from XOR in *Sod1*^−/−^ mice, we generated *Sod1*-null and XO-type- or XDH-type-knock-in (KI) double-mutant mice. Neither XO-type- nor XDH-type KI mutants altered aging-like phenotypes, such as anemia, fatty liver, muscle atrophy, and bone loss, in *Sod1*^−/−^ mice. Furthermore, allopurinol, an XO inhibitor, or apocynin, a nicotinamide adenine dinucleotide phosphate oxidase (NOX) inhibitor, failed to improve aging-like tissue degeneration and ROS accumulation in *Sod1^−/−^* mice. These results showed that XOR-mediated O_2_^•−^ production is relatively uninvolved in the age-related pathologies in *Sod1*^−/−^ mice.

## 1. Introduction

In mammalian cells, several mechanisms or pathways are associated with the production of reactive oxygen species (ROS), including superoxide (O_2_^•−^), within cells under physiological and pathological conditions. These include mitochondrial respiration, xanthine oxidoreductase (XOR), and nicotinamide adenine dinucleotide phosphate (NADPH) oxidase (NOX) [1,2]. Redox balance is physiologically maintained by the production and degradation of ROS by antioxidants, including vitamins C and E, and enzymes, such as superoxide dismutase (SOD), catalase, and glutathione peroxidase, in the cellular system.

To better understand the intracellular redox regulation, gene modification studies have been performed. Accumulating evidence has demonstrated that *Sod1*-deficient (*Sod1*^−/−^) mice show complete SOD1 protein loss and increased intracellular O_2_^•−^ as well as various aging-associated organ pathologies, such as hepatic carcinoma [3], fatty liver [4], acceleration of Alzheimer’s disease [5,6], macular degeneration [7,8], dry eye [9,10], hemolytic anemia [11], osteopenia [12,13], skin atrophy [14,15], skeletal muscle atrophy [16], luteal degeneration [17], and alteration of the gastrointestinal microbiota [18]. SOD1 loss leads to the accumulation of oxidative molecules, including lipid peroxides, carbonylated proteins, oxidized nucleic acids, and advanced glycation end products, which results in widespread impaired cellular signaling, gene expression, and cell death in tissues [19]. Therefore, SOD1 plays a central role in cytoplasmic O_2_^•−^ metabolism in intracellular redox regulation.

Mammalian XOR is ubiquitously expressed and catalyzes the conversion of hypoxanthine to xanthine and xanthine to uric acid. XOR can be found in two inter-convertible forms: (1) xanthine oxidase (XO) is an O_2_^•−^-mediated type that uses oxygen and generates O_2_^•−^, while (2) xanthine dehydrogenase (XDH) is an NAD^+^-mediated type that uses NAD^+^ as a cofactor and leads to reduced nicotinamide adenine dinucleotide production but not the generation of O_2_^•−^ [20]. These two types of XOR differ in the structure of the active site loop and the loop containing flavin adenine dinucleotide and molybdenum domains [21]. Pharmacological intervention with XO inhibitors has shown that XO is involved in various acute-injury models, such as ischemia-reperfusion injury [22,23], hyperglycemic cardiomyopathy [24], and neurodegeneration induced by spinal cord injury [25]. These results suggest that XO-mediated O_2_^•−^ production impairs organ integrity under pathological conditions. Since XOR-knockout mice die within six weeks after birth due to renal failure [26], it is difficult to elucidate the role of XOR in vivo. To clarify the pathophysiological contribution of XOR, Kusano et al. generated two types of knock-in (KI) mice for XO-locked- or XDH-stable KI mutations [21]. The XO-locked-type mice that generate O_2_^•−^, but not the XDH-stable-type mice that do not generate O_2_^•−^, showed markedly increased tumor growth associated with the activation of macrophages [21]. However, the distinct roles of XO or XDH in aging-like pathologies induced by SOD1 deficiency remain unclear.

In the present study, to investigate the pathological significance of XOR-mediated O_2_^•−^ in *Sod1*^−/−^ mice, we generated SOD1 and XO-locked type- or XDH-stable-type KI double-mutant mice and investigated the pathological association between XO-produced O_2_^•−^ and age-related pathologies caused by SOD1 deficiency. Furthermore, we administered XOR or NOX inhibitors to *Sod1*-deficient mice and investigated the protective effect of suppression of XO- and NOX-derived O_2_^•−^ on the aging-like phenotypes induced by oxidative stress.

## 2. Results

### 2.1. XO-Locked or XDH-Stable Types Failed to Improve the Aging-Like Phenotypes of Sod1^−/−^ Mice

Since the physiological and pathological roles of XO/XDH conversion remain controversial, we used XO-locked-type mutant mice (W338A/F339L) to elucidate the pathological role in XO-derived O_2_^•−^ in vivo [21,27]. To confirm the contribution of XO to the phenotypes of *Sod1^−/−^* mice, we generated *Sod1^−/−^* XO-locked-type double-mutant mice (Figure 1). *Sod1^−/−^* XO-locked-type mice were born according to the Mendelian rule without lethality, showing no apparent growth abnormalities (data not shown). As with *Sod1*^−/−^ mice, the *Sod1^−/−^* XO-locked-type double-mutant mice also showed reduced red blood cells and splenomegaly (Figure 1A,B). The XO-locked type did not further exacerbate the progression of fatty liver, muscle atrophy, bone loss, or body weight in *Sod1*^−/−^ mice (Figure 1C–F).

The XDH-stable-type mutation cannot convert to an XO form, which produces O_2_^•−^. We, therefore, expected XDH-stable-type mutations to rescue the aging-like phenotypes via reduction of the O_2_^•−^-induced oxidative damage in *Sod1*^−/−^ mice. To further clarify the protective effect of XDH against aging-like pathologies, we also analyzed the tissue changes in *Sod1*^−/−^ XDH-stable-type (C995R) KI double-mutant mice (Figure 2). *Sod1^−/−^* XDH-stable-type double-mutant mice were born normally without any lethal or apparent growth abnormalities (data not shown). Unexpectedly, XDH-stable type did not improve the hemolytic anemia associated with the splenomegaly induced by SOD1 deficiency (Figure 2A,B). Consistent with the XO-locked type, the XDH-stable type also failed to change the systemic pathologies, including fatty liver, muscle atrophy, bone loss, and low body weight in *Sod1^−/−^* mice (Figure 2C–F). These results suggest that neither XO nor XDH types of XOR are involved in the various aging-like pathologies induced by SOD1 deficiency.

### 2.2. XO Inhibitor Fails to Improve the Aging-Like Pathologies in Sod1^−/−^ Mice

In vitro studies using rodents have revealed that the administration of allopurinol (30–50 mg/kg/day), an XO inhibitor, via drinking water reduces serum uric acid by 50–90% for 2–14 weeks [28,29,30]. According to these experimental protocols, we administered allopurinol (30 mg/kg/day) for 8 weeks to determine the improvement effect of XO-derived O_2_^•−^ inhibition on tissue degeneration in *Sod1*^−/−^ mice. In wild-type (WT) mice, the administration of allopurinol did not change the body weight, suggesting no noticeable adverse effects (Figure 3A). *Sod1*^−/−^ mice exhibited no significant changes in body weight due to allopurinol administration, and their body weight remained lower than that of WT mice (Figure 3A). Unexpectedly, *Sod1*^−/−^ mice administered allopurinol exhibited significant bone loss, anemia, fatty liver, and muscle and skin atrophy compared with WT mice (Figure 3B–F).

To investigate the potential effects of sex differences, we also administered allopurinol to *Sod1*^−/−^ male mice. As in female mice, allopurinol did not improve any of these pathologies in *Sod1*^−/−^ male mice (Figure 4). Furthermore, to investigate the contribution of O_2_^•−^ derived from other pathways, we also administered apocynin (0.4 g/kg/day), an NOX inhibitor, to *Sod1*^−/−^ mice. Apocynin treatment also failed to improve the systemic aging pathologies in *Sod1*^−/−^ male mice (Figure 4). These results suggest that O_2_^•−^ derived from XO and NOX has no association with tissue pathologies in *Sod1*^−/−^ mice.

## 3. Discussion

### 3.1. Contribution of XO-Derived O_2_^•−^ to Tissue Pathology

The two types of KI mice for XO-locked- or XDH-stable TKI mutations are powerful tools for clarifying the pathological effects of XOR in various tissues. In contrast to the therapeutic effects of allopurinol on the tissue pathologies induced by acute injuries [22,23,24,25], modulation of XO activity by genetic engineering or pharmacological techniques failed to attenuate aging-associated pathologies induced by *Sod1*^−/−^ mice (Figure 1, Figure 2, Figure 3 and Figure 4). We also found that apocynin did not alter the tissue pathologies in *Sod1*^−/−^ mice (Figure 4). In an in vitro study, treatment with a mixture of allopurinol, apocynin, and N^ω^-Nitro-L-arginine methyl ester hydrochloride, an NOS inhibitor, failed to attenuate the ROS accumulation in *Sod1*^−/−^ cells (data not shown). These data strongly suggest that SOD1 does not physiologically catalyze O_2_^•−^ derived from XO, NOX, or NOS. Mitochondria produce ROS, including O_2_^•−^, through the electron transport chains of complexes I and III and release O_2_^•−^ to both sides of the inner mitochondrial membrane [31]. SOD1 is also slightly localized in the intermembrane space of mitochondria in rats and yeast [32,33,34], suggesting that SOD1 mainly catalyzes O_2_^•−^ in the intermembrane space and cytoplasm. In contrast, mitochondrial SOD2 mainly degrades O_2_^•−^ in the mitochondrial matrix. Paraquat generates mitochondrial O_2_^•−^ via complex I inhibition, resulting in mitochondrial dysfunction [35]. Paraquat treatment actually shortened the life span of *Sod1*^−/−^ mice [36], suggesting that O_2_^•−^ derived from mitochondria plays a major role in SOD1-mediated metabolism in cytoplasm. Other mechanisms of ROS production have also been reported, such as via cyclooxygenase, Fenton, and Haber–Weiss reactions mainly generating peroxy and hydroxy radicals [37]. These reactions may contribute slightly but not markedly to SOD1-mediated metabolism in cells.

SOD1 deficiency also increases ROS, proinflammatory cytokines, and lipoperoxides in various organs, including the muscle, skin, liver, and blood [4,15,19,38,39,40,41]. The status of redox, inflammation, and lipoperoxides in *Sod1*^−/−^ XOR double-mutant mice should be clarified in future studies. Furthermore, *Sod1*^−/−^ mice exhibited various tissue pathologies, including anemia, fatty liver, muscle atrophy, bone loss, and skin atrophy (Figure 1, Figure 2, Figure 3 and Figure 4). In contrast, tissue pathologies induced by XO activation occur, especially in the heart, vascular tissue, and nerves [22,23,24,25], and do not overlap with affected tissues in *Sod1*^−/−^ mice, suggesting that XO-mediated oxidative damages may show distinct organ selectivity from *Sod1*^−/−^ mice. We recently demonstrated that *Sod1* loss activates the Forkhead box O3–matrix metalloproteinase-2 axis in the skin [42], suggesting that selective signaling pathways are activated by SOD1-catalyzing O_2_^•−^ from mitochondria.

In the present study, we were unable to match the age of *Sod1*^−/−^ mice in Figure 1, Figure 2, Figure 3 and Figure 4. Although we noted no marked difference in the pathological features of *Sod1*^−/−^ mice between 4 and 12 months of age, the results need to be reinvestigated with the same protocols. Since complete SOD1 loss has not been reported in human diseases, the interpretation of our result using knockout mice is limited. A partial inhibition model of SOD1, such as that using an inhibitor, would be valuable for revealing the cross talk between aging-like pathology and O_2_^•−^ generation.

### 3.2. Contribution of XO-Derived O_2_^−^ to Tissue Pathology

XOR catalyzes the reaction steps from hypoxanthine to xanthine and from xanthine to uric acid in the pathway of purine metabolism [20]. Uric acid is also known as the classical radical scavenger, such as singlet oxygen, peroxyl radicals, and hydroxyl radicals [43,44]. Antioxidant activity of uric acid protects the erythrocyte membrane from lipid oxidation [45,46]. Uric acid also improves various pathologies, including sclerosis, Parkinson’s disease, acute stroke, ischemia-induced brain injury, allergic encephalomyelitis, doxorubicin-induced cardiotoxicity, and hepatopathy induced by hemorrhagic shock [47,48,49,50,51,52,53]. Uric acid itself also generates some radicals and acts as a pro-oxidant [43]. Uric acid amplified the oxidation of liposomes via peroxynitrite generation [54]. This oxidant–antioxidant paradox of uric acid further complicates our understanding of the contribution of XO to the tissue pathology caused by oxidative stress in *Sod1*^−/−^ mice.

### 3.3. Pathological Effect of XO on Aging and Tumorigenesis

Aging is also associated with the progressive impairment of homeostasis as a result of chronic redox imbalance and inflammation. XO is upregulated by proinflammatory cytokines, such as tumor necrosis factor-α and interleukin-6 [55]. Indeed, the XO expression and activity have been shown to increase in an age-dependent manner in the liver, kidney, thymus, aorta, and plasma [56,57,58]. XO-derived O_2_^•−^ stimulates activator protein 1 activity via c-jun N-terminal kinase and p38 in vascular smooth muscle [59], suggesting XO as a possible O_2_^•−^ donor in inflammation-related pathologies. These reports suggest that O_2_^•−^ released from mitochondria causes chronic pathologies, and XO-derived O_2_^•−^ induces acute tissue damage. 

A high level of ROS leads to the activation of various oncogenic pathways [60]. We previously demonstrated that XO-locked-type KI mice promote tumor growth due to increased ROS production in macrophages [21]. *Sod1*^−/−^ mice also show hepatocarcinogenesis progress associated with the accumulation of oxidative damage in late life stages [3]. To clarify the contribution of SOD1 to tumorigenesis, we generated *Sod1*^−/−^
*p53*^−/−^ double-knockout mice. Compared to *p53*^−/−^ mice, *Sod1*^−/−^
*p53*^−/−^ double-knockout mice showed significantly accelerated tumorigenesis by 4 months of age (Watanabe et al. submitted), suggesting alteration of the tumorigenesis profiles by redox imbalance. Neither *Sod1*^−/−^ XO-locked-type nor *Sod1*^−/−^ XDH-stable-type double-mutant mice exhibited tumor formation in the liver or other tissues at 7–12 months of age (data not shown). At older ages, XO-locked- and XDH-stable-type mutations may alter tumorigenesis in *Sod1*^−/−^ mice. To clarify the contribution of XOR to tumorigenesis in *Sod1*^−/−^ mice, further analyses are needed to characterize the pathogenic macrophages and tumorigenesis in *Sod1*^−/−^ XO-locked-type and *Sod1*^−/−^ XDH-stable-type double-mutant mice.

In conclusion, KI mutations of XOR and inhibitors of XO or NOX did not alter the aging-like pathologies in *Sod1*^−/−^ mice, suggesting that XOR-mediated O_2_^•−^ production contributes relatively little to the aging-like pathologies in *Sod1*^−/−^ mice. SOD1 may use O_2_^•−^ produced by physiological and biological systems, such as mitochondrial energy production. The production source of O_2_^•−^ may affect tissue homeostasis and the optimal therapeutic strategy in ROS-related diseases in humans. Our results provide new insight into the pathophysiological role of O_2_^•−^ in oxygen metabolism by SOD and XOR.

## 4. Materials and Methods

### 4.1. Animals and Genotyping

*Sod1*^−/−^ mice were purchased from the Jackson Laboratory (Bar Harbor, ME, USA). Genotyping of the *Sod1*^−/−^ allele was performed via genomic polymerase chain reaction using genomic DNA isolated from the tail tip, as reported previously [14]. Two Xdh gene-modified mice, the XO-locked (W338A/F339L mutant) and the XDH-stable (C995R mutant) types [21], were kindly provided by Drs. Teruo Kusano, Takeshi Nishino (Nippon Medical School, Bunkyo-ku, Japan), and Ken Okamoto (The University of Tokyo, Bunkyo-ku, Japan). These animals were housed under a 12 h/12 h light/dark cycle and fed ad libitum. In addition, they were maintained and studied according to the protocols approved by the animal care committee of Chiba University and the National Center for Geriatrics and Gerontology.

### 4.2. Administration of Allopurinol and Apocynin

The WT and *Sod1*^−/−^ mice were given 1 mM allopurinol (30 mg/kg/day, 09-12502; FUJIFILM Wako Pure Chemical Corporation, Osaka, Japan) and 2 mg/mL of apocynin (0.4 g/ kg/day, 6002; EXTRASYNTHASE, Metropole de Lyon, France) in drinking water daily for 8 weeks from 7 weeks of age (female) and 12 weeks of age (male).

### 4.3. Analysis of Aging-Like Pathologies

Red blood cells were measured using Celltac a (MEK-6358; NIHON KODEN, Shinjuku-ku, Japan). Bone mineral density of the whole body was measured using a PIXImus instrument (Lunar Corp., Madison, WI, USA). The thickness of the isolated back skin was measured using a PEACOCK dial thickness gauge (OZAKI MFG. CO., LTD., Itabashi-ku, Japan).

### 4.4. Statistical Analyses

Statistical analyses were performed using one-way analysis of variance/Tukey’s test for comparisons of three or more groups. Differences between data were considered significant when *p*-values were less than 0.05. All data are expressed as the mean ± standard deviation (SD).

## Figures and Tables

**Figure 1 ijms-22-03542-f001:**
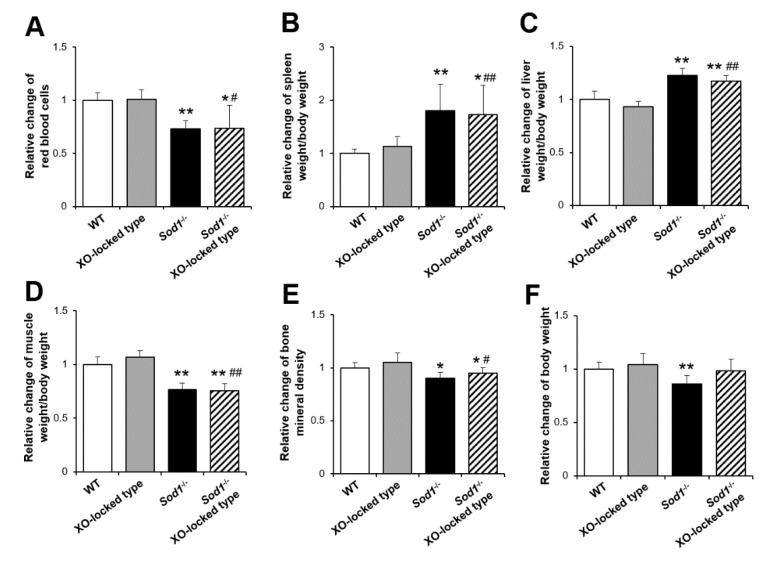
A phenotypical analysis of *Sod1*, xanthine oxidase (XO)-locked-type double-mutant mice. Relative changes in the red blood cell count (**A**), ratio of spleen weight/body weight (**B**), ratio of liver weight/body weight (**C**), ratio of muscle weight/body weight (**D**), bone mineral density (**E**), and body weight (**F**) of wild-type (WT), *Sod1*^−/−^, XO-locked^-^type, and *Sod1*^−/−^ XO-locked-type male and female mice (4–5 months of age, *n* = 5–6). * *p* < 0.05, ** *p* < 0.01 vs. WT. ^#^
*p* < 0.05, ^##^
*p* < 0.01 vs. XO-locked type. Data are shown as the mean ± SD.

**Figure 2 ijms-22-03542-f002:**
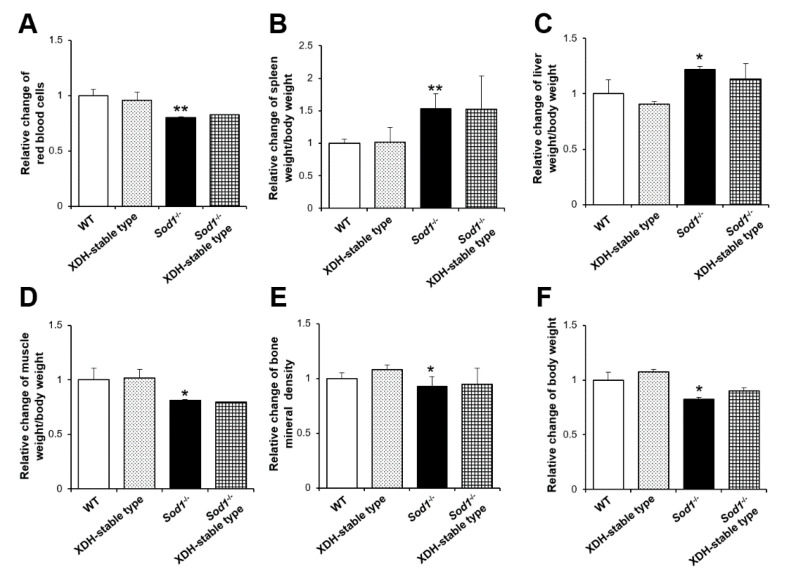
A phenotypical analysis of *Sod1*, XDH-stable-type double-mutant mice. Relative changes in the red blood cell count (**A**), spleen weight (**B**), ratio of spleen weight/body weight (**C**), ratio of muscle weight/body weight (**D**), bone mineral density (**E**), and body weight (**F**) of WT, *Sod1*^−/−^, XDH-stable, and *Sod1*^−/−^ XDH-stable male mice (4–5 months of age, *n* = 3). * *p* < 0.05, ** *p* < 0.01. Data are shown as the mean ± SD.

**Figure 3 ijms-22-03542-f003:**
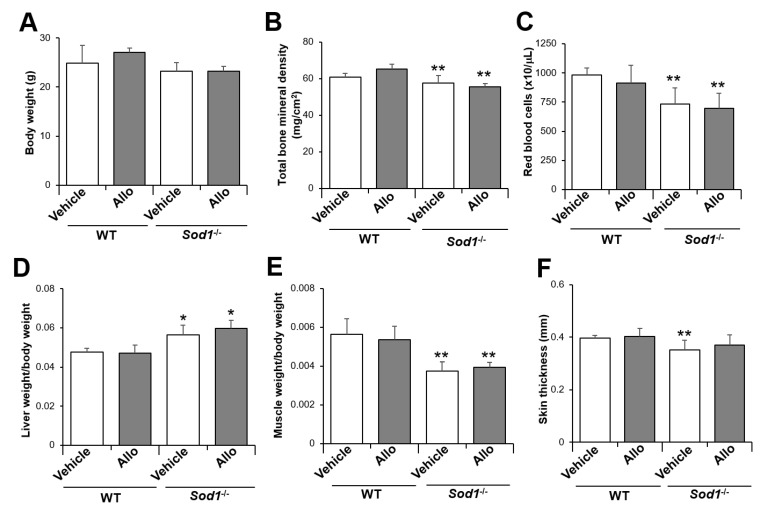
Administration of allopurinol to adult *Sod1*^−/−^ female mice. The body weight (**A**), total bone mineral density (**B**), red blood cell count (**C**), ratio of liver weight/body weight (**D**), ratio of muscle weight/body weight (**E**), and skin thickness (**F**) of WT and *Sod1*^−/−^ female mice (7 months of age, *n* = 4–5) administered allopurinol for 8 weeks. * *p* < 0.05, ** *p* < 0.01. Data are shown as the mean ± SD.

**Figure 4 ijms-22-03542-f004:**
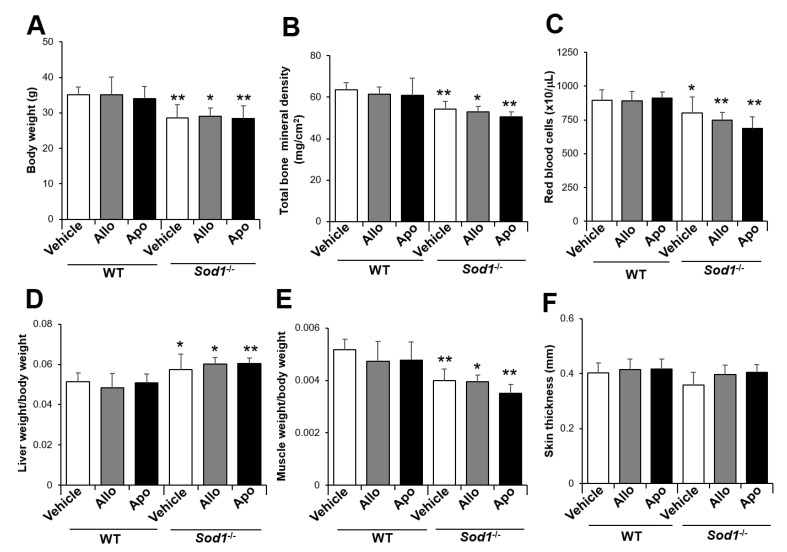
Administration of allopurinol to aged *Sod1*^−/−^ male mice. The body weight (**A**), total bone mineral density (**B**), red blood cell count (**C**), ratio of liver weight/body weight (**D**), ratio of muscle weight/body weight (**E**), and skin thickness (**F**) of WT and *Sod1*^−/−^ male mice (12 months of age, *n* = 4–5) administered allopurinol or apocynin for 8 weeks. * *p* < 0.05, ** *p* < 0.01. Data are shown as the mean ± SD.

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
