# Peer review of "Xanthine Oxidoreductase-Mediated Superoxide Production Is Not Involved in the Age-Related Pathologies in *Sod1*-Deficient Mice"

_ijms, 2021, doi:10.3390/ijms22073542_

Round 1
Reviewer 1 Report
The manuscript presents results of a study on ROS in ageing processes in transgenic mice.
It is usually difficult to publish negative results, and here the Authors' hypothesis of potential effect was rejected. However such studies are also very important.
Title suggest the existing effect, therefore it is misleading - please change that to more appropriate.
Author Response
The manuscript presents results of a study on ROS in ageing processes in transgenic mice.
It is usually difficult to publish negative results, and here the Authors' hypothesis of potential effect was rejected. However, such studies are also very important.
Title suggest the existing effect, therefore it is misleading - please change that to more appropriate.
Response: As suggested, we have revised the title of this paper as follows: ‘Xanthine oxidoreductase-mediated superoxide production is not involved in the age-related pathologies of Sod1-deficient mice’

Reviewer 2 Report
The manuscript “Pathological significance of xanthine oxidoreductase-mediated superoxide generation in Sod1-deficient mice” shows results which indicate that XO/XDH has no role in Sod-/- induced oxidative stress during aging.
The paper is well written but there are some points to be clarified. The title is misleading.
Introduction
XO/XDH, please explain the inter-convertible conditions.
Clarify the aim of the paper. (Especially in the Abstract).
Results
In Fig. 3 and 4. Gender should be indicated in the title of thee figure (the first sentence explaining briefly the figure).
Explain the difference in age of male and female mice and why the two gender did not follow the same protocol. The males were 12 month and females were 7 month of age, the treatment did not start at the same age and these factors could contribute to results.
Figures should be repositioned after the section which explains them, currently it is a bit confusing to follow the figures and the text of the Results.
It is a bit confusing why authors did ROS intensity in Sod-/- fibroblasts. The point is already shown with animals.
Discussion
“These data strongly suggest that SOD1 physiologically catalyzes O2•- derived from mitochondria or NOX but not XO.” How is it, if the authors showed that: “Apocynin treatment also failed to improve the systemic aging pathologies of Sod1-/- male mice (Figure 4).”
Authors state that Sod1 catalyzes degradation of mitochondrial O2-, what about Sod2. There are also other sources of ROS in the cytoplasm, as Cox and Fenton and Haber-Wiess reaction of transition metals.
The authors cite paper which is submitted, not accepted, please change this. Add the tumor issue in the Introduction in order to provide link between Introduction and Discussion.
Materials and methods
There should be some explanation for concentrations of allopurinol, apocynin and L-NAME used in the paper.
Indicate time of exposure for ROS measurement in fibroblasts. Provide explanation for concentrations of inhibitors used for ROS measurement. Also, why the authors did no provide viability data for the inhibitors?
Author Response
The manuscript “Pathological significance of xanthine oxidoreductase-mediated superoxide generation in Sod1-deficient mice” shows results which indicate that XO/XDH has no role in Sod-/- induced oxidative stress during aging.
The paper is well written but there are some points to be clarified. The title is misleading.
Response: As suggested, we have revised the title of this paper as follows: ‘Xanthine oxidoreductase-mediated superoxide production is not involved in the age-related pathologies of Sod1-deficient mice’
Introduction
XO/XDH, please explain the inter-convertible conditions.
Response: As suggested, we have added detailed descriptions of the inter-convertible conditions of XO/XDH to the ‘Introduction’.
(Lines 52-54 in Introduction) ‘These two types of XOR differ in the structure of the active site loop and the loop containing flavin adenine dinucleotide and molybdenum domains [21]’
Clarify the aim of the paper. (Especially in the Abstract).
Response: As suggested, we have clearly described the aim of this study in the ‘Abstract’ and ‘Introduction’.
(Lines 18-20 in Abstract) ‘In order to investigate the pathological significance of O2•- derived from XOR in Sod1-/- mice, we generated Sod1 null and XO type- or XDH type-knock-in (KI) double-mutant mice.’
(Lines 66-69 in Introduction) ‘In the present study, to investigate the pathological significance of XOR-mediated O2- in Sod1-/- mice, we generated SOD1 and XO-locked type- or XDH-stable type-KI double-mutant mice and investigated the pathological association between XO-produced O2•- and age-related pathologies caused by SOD1 deficiency.’
Results
In Fig. 3 and 4. Gender should be indicated in the title of the figure (the first sentence explaining briefly the figure).
Response: As suggested, we have described the gender in the legends of Figures 3 and 4.
Explain the difference in age of male and female mice and why the two gender did not follow the same protocol. The males were 12 month and females were 7 months of age, the treatment did not start at the same age and these factors could contribute to results.
Response: We apologize for our insufficient description of the experiment setting. In this paper, we were unable to match the ages of the mice in each experiment. We have now mentioned this in the ‘Discussion’.
(Lines 169-172, in Discussion) ‘In the present study, we were unable to match the age of Sod1-/- mice in Figures 1-4. Although we noted no marked difference in the pathological features of Sod1-/- mice between 4 and 12 months of age, the results need to be reinvestigated with the same protocols.’
Figures should be repositioned after the section which explains them, currently it is a bit confusing to follow the figures and the text of the Results.
Response: We apologize for the confusion. As suggested, we have reconstructed the arrangement of the figures in the ‘Results’ section.
It is a bit confusing why authors did ROS intensity in Sod-/- fibroblasts. The point is already shown with animals.
Response: As suggested, we deleted ‘Figure 5’ for the in vitro study.
Discussion
“These data strongly suggest that SOD1 physiologically catalyzes O2•- derived from mitochondria or NOX but not XO.” How is it, if the authors showed that: “Apocynin treatment also failed to improve the systemic aging pathologies of Sod1-/- male mice (Figure 4).”
Response: To avoid confusion, we have revised the indicated description.
(Lines 144-146 in Discussion) ‘These data strongly suggest that SOD1 does not physiologically catalyze O2•- derived from XO, NOX, or NOS.’
Authors state that Sod1 catalyzes degradation of mitochondrial O2-, what about Sod2. There are also other sources of ROS in the cytoplasm, as Cox and Fenton and Haber-Wiess reaction of transition metals.
Response: Since the ROS generated by cyclooxygenase, Fenton and Haber-Wiess reaction are mainly peroxy and hydroxy radicals, they are not considered to significantly influence the phenotypes of Sod1-/- mice. We have now added a relevant reference and description of these other ROS sources to the ‘Discussion’.
(Lines 154-157 in Discussion) ‘Other mechanisms of ROS production have also been reported, such as via cyclooxygenase, Fenton and Haber-Weiss reactions mainly generating peroxy and hydroxy radicals, [37]. These reactions may contribute slightly but not markedly to SOD1-mediated metabolism in cells.’
References
[37] Snezhkina, A.V.; Kudryavtseva, A.V.; Kardymon, O.L.; Savvateeva, M.V.; Melnikova, N.V.; Krasnov, G.S.; Dmitriev, A.A. ROS Generation and Antioxidant Defense Systems in Normal and Malignant Cells. Oxid Med Cell Longev 2019, 2019, 6175804.
The authors cite paper which is submitted, not accepted, please change this.
Response: The latest paper (Nashi et al. 2021) we referenced was not reflected in PubMed at the time of the submission of this article. However, the paper is now available in PubMed.
[27] Nasi, S.; Castelblanco, M.; Chobaz, V.; Ehirchios, D.; So, A.; Bernabei, I.; Kusano, T.; Nishino, T.; Okamoto, K.; Busso, N. Xanthine oxidoreductase Is involved in chondrocyte mineralization and expressed in osteoarthritic damaged cartilage. Front Cell Dev Biol 2021, 9, 612440.
Add the tumor issue in the Introduction in order to provide link between Introduction and Discussion.
Response: We have now described the tumor in the ‘Introduction’ to this paper.
(Lines 62-64 in Introduction) ‘The XO-locked-type mice that generate O2•-, but not the XDH-stable type that does not generate O2•-, show markedly increased tumor growth associated with the activation of macrophages [21].’
Materials and methods
There should be some explanation for concentrations of allopurinol, apocynin and L-NAME used in the paper.
Indicate time of exposure for ROS measurement in fibroblasts. Provide explanation for concentrations of inhibitors used for ROS measurement. Also, why the authors did no provide viability data for the inhibitors?
Response: Because we deleted Figure 5, we also omitted the description of Figure 5 in the ‘Materials and Methods’.

Reviewer 3 Report
In the present study, was analyzed the effect of SOD1 and XO-locked type- or XDH-stable type- KI double-mutant mice and age-related pathologies caused by SOD1 deficiency. The authors demonstrated that KI mutations of XOR and inhibitors of XO or NOX did not alter the aging-like pathologies in Sod1-/- mice. However I have some curiosities
- Have the authors consiedered the possibility to evaluate the plasmatic antioxidant/oxidant status in mice
- In addition could the authors evaluate the lipoperoxidation status of muscle, liver, skin and red blood cells
- Have you also thought of evaluate the blood inflammatory status in the experimental groups
If you cannot perform these experiments, are there already published data that you could include in the discussion section?
Author Response
In the present study, was analyzed the effect of SOD1 and XO-locked type- or XDH-stable type- KI double-mutant mice and age-related pathologies caused by SOD1 deficiency. The authors demonstrated that KI mutations of XOR and inhibitors of XO or NOX did not alter the aging-like pathologies in Sod1-/- mice. However, I have some curiosities
Have the authors considered the possibility to evaluate the plasmatic antioxidant/oxidant status in mice.
In addition, could the authors evaluate the lipoperoxidation status of muscle, liver, skin and red blood cells.
Have you also thought of evaluate the blood inflammatory status in the experimental groups.
If you cannot perform these experiments, are there already published data that you could include in the discussion section?
Response: We also believe that the inclusion of additional data, including the status of redox, lipoperoxides, and inflammation, is very important. Unfortunately, we were unable to perform this experiment due to the 10-day revision period. Instead, to augment the data, we have added references and descriptions of these statuses in Sod1-/- mice.
(Lines 158-161 in Discussion) ‘SOD1 deficiency also increases ROS, proinflammatory cytokines, and lipoperoxides in various organs, including the muscle, skin, liver, and blood [4,19,38-42]. The status of redox, inflammation, and lipoperoxides in Sod1-/-, XOR double mutant mice should be clarified in future studies.’
References
[4] Uchiyama, S.; Shimizu, T.; Shirasawa, T. CuZn-SOD deficiency causes ApoB degradation and induces hepatic lipid accumulation by impaired lipoprotein secretion in mice. J Biol Chem 2006, 281, 31713-31719.
[19] Watanabe, K.; Shibuya, S.; Ozawa, Y.; Nojiri, H.; Izuo, N.; Yokote, K.; Shimizu, T. Superoxide dismutase 1 loss disturbs intracellular redox signaling, resulting in global age-related pathological changes. Biomed Res Int 2014, 2014, 140165.
[38] Shibuya, S.; Ozawa, Y.; Toda, T.; Watanabe, K.; Tometsuka, C.; Ogura, T.; Koyama, Y.; Shimizu, T. Collagen peptide and vitamin C additively attenuate age-related skin atrophy in Sod1-deficient mice. Biosci Biotechnol Biochem 2014, 78, 1212-1220.
[39] Shibuya, S.; Ozawa, Y.; Watanabe, K.; Izuo, N.; Toda, T.; Yokote, K.; Shimizu, T. Palladium and platinum nanoparticles attenuate aging-like skin atrophy via antioxidant activity in mice. PLoS One 2014, 9, e109288.
[40] Qaisar, R.; Bhaskaran, S.; Ranjit, R.; Sataranatarajan, K.; Premkumar, P.; Huseman, K.; Van Remmen, H. Restoration of SERCA ATPase prevents oxidative stress-related muscle atrophy and weakness.
[41] Jang, Y.C.; Rodriguez, K.; Lustgarten, M.S.; Muller, F.L.; Bhattacharya, A.; Pierce, A.; Choi, J.J.; Lee, N.H.; Chaudhuri, A.; Richardson, A.G., et al. Superoxide-mediated oxidative stress accelerates skeletal muscle atrophy by synchronous activation of proteolytic systems. Geroscience 2020, 42, 1579-1591.
[42] Bhaskaran, S.; Pollock, N.; P, C.M.; Ahn, B.; Piekarz, K.M.; Staunton, C.A.; Brown, J.L.; Qaisar, R.; Vasilaki, A.; Richardson, A., et al. Neuron-specific deletion of CuZnSOD leads to an advanced sarcopenic phenotype in older mice. Aging Cell 2020, 19, e13225.

Reviewer 4 Report
This is a very interesting study investigating the crosstalk between SOD1 and XOR using genetically modified animal models, and its impact on age-related disorders triggered by SOD1 deletion. Nevertheless, the authors have chosen an extreme experimental condition (SOD-/-) which makes difficult the investigation of the crosstalk with other physiological superoxide-generating mechanisms. Would it be more relevant to inhibit (partially) SOD1 activity (for instance by using LCS-1)? In my opinion the limitations of the study should be mentioned in the Discussion section.
The presentation is clear, with some amendments that I have made below.
Abstract
- Explain briefly what XDH is doing, as you investigated XDH in the study;
- At the end, please briefly comment on the significance of results from a network biology perspective, and their potential impact of your results in humans.
Results
- Lines 87-98: Mention the figure number after each type of SOD-/- pathology
- Figures 1-4: specify the number of mice in each group.
- Figure 5: SD bars are needed as well as comparison between various conditions, otherwise it is impossible to establish the effect of allopurinol and of the inhibitors cocktail on WT and SOD1-/- fibroblasts.
- There are differences in the age of mice in various studies (4-5 months, 7 months, 12 months). Is this difference in mice age influencing the correlation of data from various experiments? How are evolving with age the pathological features of SOD-/- mice as well as of transgenic mice? At least in WT mice you cannot consider mice with the age 4-5 months to be old. Apparently, by deleting the SOD1 gene you force the emergence of old-age pathologies at lower age (4-5 months).
- Do you have proof that you inhibited XO (and to what extent) by treating mice with allopurinol?
Discussion
- Lines 152-153: In my opinion, your results show that deletion of the SOD1 gene triggers massive production of superoxide anion with important pathological consequences. The amount of superoxide anion generated through the XO pathway is far lower and, therefore, the genetic or pharmacological modulation of the XOR-mediated pathway cannot significantly influence the pathologies inflicted by the absence of SOD1. In the same way we may comment on the contribution of NOX and NOS in SOD1-/- related pathologies. Nevertheless, results show the importance of SOD1 in generating some old-age conditions that you have analyzed in the study.
- Lines 153-158: could it be possible that mitochondrial generation of superoxide using paraquat adds to the superoxide accumulation due to SOD1 deletion, and affects therefore the life span of SOD1-/- mice? To my knowledge, only a small amount of SOD1 localizes in the mitochondria, and may be directed to the nucleus in response to increased level of H2O2 to prevent oxidative genomic damage (https://doi.org/10.1038/ncomms4446).
- Lines 200-204: Please reconsider the conclusion of the study. This has to be mentioned also in the abstract.
Methods
- Lines 217-220: mention the dosage as mg/body weight. Was treatment provided on a daily basis?
Author Response
This is a very interesting study investigating the crosstalk between SOD1 and XOR using genetically modified animal models, and its impact on age-related disorders triggered by SOD1 deletion. Nevertheless, the authors have chosen an extreme experimental condition (SOD-/-) which makes difficult the investigation of the crosstalk with other physiological superoxide-generating mechanisms. Would it be more relevant to inhibit (partially) SOD1 activity (for instance by using LCS-1)? In my opinion the limitations of the study should be mentioned in the Discussion section.
Response: As suggested, we have described the limitations of this study and the need for SOD1 inhibitors in the ‘Discussion’.
(Lines 172-175 in Discussion) ‘Since complete SOD1 loss has not been reported in human diseases, the interpretation of our result using knockout mice is limited. A partial inhibition model of SOD1, such as that using an inhibitor, would be valuable for revealing the crosstalk between aging-like pathology and O2•- generation.’
The presentation is clear, with some amendments that I have made below.
Abstract
Explain briefly what XDH is doing, as you investigated XDH in the study;
Response: Due to the character limit, we were unable to add more description about XDH to the ‘Abstract’. Instead, we have clearly described the aim of this study in the ‘Abstract’.
(Lines 18-20 in Abstract) ‘In order to investigate the pathological significance of O2•- derived from XOR in Sod1-/- mice, we generated Sod1 null and XO type- or XDH type-knock-in (KI) double-mutant mice.’
At the end, please briefly comment on the significance of results from a network biology perspective, and their potential impact of your results in humans.
Response: Due to the character limit, we were unable to add more description to the ‘Abstract’. Instead, we have described the application of our findings to humans to the ‘Discussion’.
(Lines 217-218, in Discussion) ‘The production source of O2•- may affect tissue homeostasis and the optimal therapeutic strategy in ROS-related diseases in humans.’
Results
Lines 87-98: Mention the figure number after each type of SOD-/- pathology
Response: As suggested, we have added appropriate figure numbers to the text.
Figures 1-4: specify the number of mice in each group.
Response: As suggested, we have added the number of mice in each group to the figure legends.
Figure 5: SD bars are needed as well as comparison between various conditions, otherwise it is impossible to establish the effect of allopurinol and of the inhibitors cocktail on WT and SOD1-/- fibroblasts.
Response: Because the experimental conditions were insufficient, we deleted ‘Figure 5’ for the in vitro study.
There are differences in the age of mice in various studies (4-5 months, 7 months, 12 months). Is this difference in mice age influencing the correlation of data from various experiments? How are evolving with age the pathological features of SOD-/- mice as well as of transgenic mice? At least in WT mice you cannot consider mice with the age 4-5 months to be old. Apparently, by deleting the SOD1 gene you force the emergence of old-age pathologies at lower age (4-5 months).
Response: We apologize for our insufficient description of the experiment setting. In this paper, we were unable to match the ages of the mice in each experiment. We have now mentioned this in the ‘Discussion’.
(Lines 169-172, in Discussion) ‘In the present study, we were unable to match the age of Sod1-/- mice in Figures 1-4. Although we noted no marked difference in the pathological features of Sod1-/- mice between 4 and 12 months of age, the results need to be reinvestigated with the same protocols.’
Do you have proof that you inhibited XO (and to what extent) by treating mice with allopurinol?
Response: In previous reports, the administration of 30-50 mg/kg/day allopurinol via drinking water reduced serum uric acid by 50%-90% at 2-14 weeks. With reference to these reports, we administered allopurinol to mice (30 mg/kg/day) for 8 weeks. We have now added references and descriptions of allopurinol treatment to the ‘Results’.
(Lines 107-111 in Results) ‘In vitro studies using rodents revealed that the administration of allopurinol (30-50 mg/kg/day), an XO inhibitor, via drinking water reduced serum uric acid by 50%-90% for 2-14 weeks [28-30]. According to these experimental protocols, we administered allopurinol (30 mg/kg/day) for 8 weeks to determine the improvement effect of XO-derived O2•- inhibition on tissue degeneration of Sod1-/- mice.’
References
[28] Kosugi, T.; Nakayama, T.; Heinig, M.; Zhang, L.; Yuzawa, Y.; Sanchez-Lozada, L.G.; Roncal, C.; Johnson, R.J.; Nakagawa, T. Effect of lowering uric acid on renal disease in the type 2 diabetic db/db mice. Am J Physiol Renal Physiol 2009, 297, F481-488.
[29] Inhibition of xanthine oxidase by allopurinol prevents skeletal muscle atrophy: role of p38 MAPKinase and E3 ubiquitin ligases.
[30] Kato, S.; Shirakami, Y.; Yamaguchi, K.; Mizutani, T.; Ideta, T.; Nakamura, H.; Ninomiya, S.; Kubota, M.; Sakai, H.; Ibuka, T., et al. Allopurinol suppresses szoxymethane-induced colorectal tumorigenesis in C57BL/KsJ-db/db mice. Gastrointestinal disorders 2020, 2, 385-396.
Discussion
Lines 152-153: In my opinion, your results show that deletion of the SOD1 gene triggers massive production of superoxide anion with important pathological consequences. The amount of superoxide anion generated through the XO pathway is far lower and, therefore, the genetic or pharmacological modulation of the XOR-mediated pathway cannot significantly influence the pathologies inflicted by the absence of SOD1. In the same way we may comment on the contribution of NOX and NOS in SOD1-/- related pathologies. Nevertheless, results show the importance of SOD1 in generating some old-age conditions that you have analyzed in the study.
Response: We have additionally mentioned the contribution of NOX and NOS as other O2•- sources for Sod1-/- mice in the ‘Discussion’.
(Lines 141-146 in Discussion) ‘We also found that apocynin did not alter the tissue pathologies of Sod1-/- mice (Figure 4). In an in vitro study, treatment with mixture of allopurinol, apocynin, and L-NAME, an NOS inhibitor, failed to attenuate the ROS accumulation in Sod1-/- cells (data not shown). These data strongly suggest that SOD1 does not physiologically catalyze O2•- derived from XO, NOX, or NOS.’
Lines 153-158: could it be possible that mitochondrial generation of superoxide using paraquat adds to the superoxide accumulation due to SOD1 deletion, and affects therefore the life span of SOD1-/- mice? To my knowledge, only a small amount of SOD1 localizes in the mitochondria, and may be directed to the nucleus in response to increased level of H2O2 to prevent oxidative genomic damage (https://doi.org/10.1038/ncomms4446).
Response: We have now added this reference and description of the localization of SOD1 to the ‘Discussion’ section.
(Lines 146-150 in Discussion) ‘Mitochondria produce ROS, including O2•-, through the electron transport chains of complexes I as well as III and release O2- to both sides of the inner mitochondria membrane [31]. SOD1 is also slightly localized in the intermembrane space of mitochondria in rats and yeast [32-34], suggesting that SOD1 mainly catalyzes O2•- in the intermembrane space and cytoplasm.’
References
[34] Tsang, C.K.; Liu, Y.; Thomas, J.; Zhang, Y.; Zheng, X.F. Superoxide dismutase 1 acts as a nuclear transcription factor to regulate oxidative stress resistance. Nat Commun 2014, 5, 3446.
Lines 200-204: Please reconsider the conclusion of the study. This has to be mentioned also in the abstract.
Response: Due to the character limit, we were unable to add more description to the ‘Abstract’. Instead, we have revised the description of the conclusion in the ‘Discussion’.
(Lines 213-219 in Discussion) ‘In conclusion, KI mutations of XOR and inhibitors of XO or NOX did not alter the aging-like pathologies in Sod1-/- mice, suggesting that XOR-mediated O2•- production contributes relatively little to the aging-like pathologies in Sod1-/- mice. SOD1 may use O2•- produced by physiological and biological systems, such as mitochondrial energy production. The production source of O2•- may affect tissue homeostasis and the optimal therapeutic strategy in ROS-related diseases in humans. Our results provide new insight into the pathophysiological role of O2•- in oxygen metabolism by SOD and XOR.’
Methods
Lines 217-220: mention the dosage as mg/body weight. Was treatment provided on a daily basis?
Response: We administered allopurinol and apocynin daily to mice through their drinking water. As suggested, we have described the administration protocol, including the dosage, in the ‘Materials and methods’ and ‘Results’.

Round 2
Reviewer 2 Report
The authors have addressed all the issues.
Reviewer 3 Report
The authors have correctly replicated according to the indications
Reviewer 4 Report
The authors answered the main issues raised and, in my opinion, the message is now clear.